# Discovery and Characterization of an ALFA-Tag-Specific Affinity Resin Optimized for Protein Purification at Low Temperatures in Physiological Buffer

**DOI:** 10.3390/biom11020269

**Published:** 2021-02-12

**Authors:** Markus Kilisch, Hansjörg Götzke, Maja Gere-Becker, Alexander Crauel, Felipe Opazo, Steffen Frey

**Affiliations:** 1NanoTag Biotechnologies GmbH, Rudolf-Wissell-Straße 28a, 37079 Göttingen, Germany; markus.kilisch@nano-tag.com (M.K.); Hansjoerg.Goetzke@nano-tag.com (H.G.); maja.Gere-Becker@nano-tag.com (M.G.-B.); Alexander.Crauel@nano-tag.com (A.C.); fopazo@gwdg.de (F.O.); 2Institute of Neuro- and Sensory Physiology, University Medical Center Göttingen, Humboldtallee 23, 37073 Göttingen, Germany; 3Center for Biostructural Imaging of Neurodegeneration (BIN), University Medical Center, Von-Siebold-Straße 3a, 37075 Göttingen, Germany

**Keywords:** nanobody, sdAbs, epitope tag, affinity, immunoprecipitation, native elution, cold-elutable, synthetic library, ALFA system, ALFA Selector

## Abstract

Epitope tags are widely employed as tools to detect, purify and manipulate proteins in various experimental systems. We recently introduced the ALFA-tag together with two ALFA-specific single-domain antibodies (sdAbs), NbALFA and NbALFA^PE^, featuring high or intermediate affinity, respectively. Together, the ALFA system can be employed for a broad range of applications in microscopy, cell biology and biochemistry requiring either extraordinarily stable binding or mild competitive elution at room temperature. In order to further enhance the versatility of the ALFA system, we, here, aimed at developing an sdAb optimized for efficient elution at low temperatures. To achieve this, we followed a stringent selection scheme tailored to the specific application. We found candidates combining a fast capture of ALFA-tagged proteins with an efficient competitive elution at 4 °C in physiological buffer. Importantly, by employing a structure-guided semisynthetic library based on well-characterized NbALFA variants, the high specificity and consistent binding of proteins harboring ALFA-tags at either terminus could be maintained. ALFA Selector^CE^, a resin presenting the cold-elutable NbALFA^CE^, is an ideal tool for the one-step purification of sensitive protein complexes or temperature-labile enzymes. We believe that the general approach followed during the selection and screening can be transferred to other challenging sdAb discovery projects.

## 1. Introduction

Camelid single-domain antibodies (sdAbs) [1], also referred to as nanobodies^®^ (a trademark of Ablynx), are employed in fundamental research and clinical diagnostics, and as promising candidates for therapeutic applications. Due to their small size of ~15 kDa, easy genetic accessibility and low production cost, they are ideal tools for many fields of life sciences [1,2,3,4,5]. Furthermore, sdAbs can be easily selected from phage-, bacterial-, or yeast-display libraries, allowing for a specific selection of binders against desired target proteins [6,7,8]. Although the selection of sdAbs from immune libraries generated from peripheral blood mononuclear cells (PBMCs) of immunized alpacas, llamas or camels is a straightforward process, provoking the required immune responses in these animals can be challenging. This is the case, in particular, for toxic or highly conserved proteins. In addition, immunizing animals is a time-consuming process with intrinsically unpredictable outcomes regarding the nature, quality and specificity of the sdAbs obtained. To overcome these limitations, some fully synthetic libraries have been developed, typically based on commonly found immunoglobulin frameworks present in camelid sdAbs [9,10]. While most sdAbs have been selected and optimized to display high affinity towards their antigens (for examples, see [11,12,13,14,15]), the generation of mid- and low-affinity sdAbs is of growing interest [16]. For instance, reversible binding to a specific target protein can be advantageous not only for in vivo and therapeutic use [16] but also for biochemical applications, enabling binding and elution under physiological conditions and thereby preserving the structure and function of selected target proteins [16,17].

Recently, we introduced the ALFA system, comprising the rationally designed ALFA-tag and a set of highly versatile sdAbs binding ALFA-tagged proteins with extraordinary specificity regardless of the position of the tag on the protein. The high-affinity sdAb (NbALFA) shows tight binding to ALFA-tagged proteins (K_d_~25 pM) with a very slow off-rate and is ideal for applications requiring stable interactions [15]. It may, however, be suboptimal for a range of biochemical applications, as it fails to release targets under physiological conditions. In order to solve this problem, we introduced a second sdAb (NbALFA^PE^ for “peptide-elutable”) showing a strongly enhanced off-rate (K_d_~11 nM). An agarose-based resin featuring immobilized NbALFA^PE^ (ALFA Selector^PE^) allows for an efficient competitive elution of ALFA-tagged proteins under physiological conditions. In summary, the ALFA system is ideally suited for a broad spectrum of applications, including high-resolution microscopy (e.g., fluorescence microscopy, STED microscopy, DNA-PAINT, etc.) [18,19,20], in vivo detection and manipulation of living cells [21,22], intracellular proteins [22,23], and also advanced biochemical experimentation [15,24]. Therefore, it offers a superior and versatile alternative to most common epitope tag systems [25].

While competitive elution from ALFA Selector^PE^ works exceptionally well in batch or stopped-flow elution protocols performed at room temperature, the elution efficiency is impaired at lower temperatures. This may limit the application of the ALFA system, especially for the purification of temperature-sensitive targets or delicate protein complexes, or in the case that it is essential to perfectly preserve structure and function. Therefore, we set out to complement the ALFA system by adding a third sdAb that allows for the efficient capture and elution of proteins at 4 °C and under physiological buffer conditions while maintaining the exquisite specificity and favorable biochemical properties of the existing NbALFA variants [15]. To achieve this, we combined a structure-guided protein engineering approach with a novel off-rate-driven phage display selection. We demonstrate that the applied rational design principles in combination with an application-specific selection protocol can lead to affinity reagents fulfilling the experimental requirements in a directed fashion. We finally present NbALFA^CE^ (for “cold-elutable”), a new member of the ALFA system, which proves to be an ideal tool for the purification and elution of ALFA-tagged target proteins at cold temperatures in physiological buffer.

## 2. Materials and Methods

### 2.1. Animal Handling—Immunizations

All the work involving animal experiments was conducted in compliance with ethical regulations for animal research and testing. All experiments conducted did not require ethical approval, but were communicated to and accepted by the local authorities (LAVES, Niedersachsen, Germany). Two alpacas were immunized six times at 14-day intervals with a total of 0.5 mg of ALFA peptide conjugated to keyhole limpet hemocyanin (KLH). The first immunization was performed using complete Freund’s adjuvant; for all the following immunizations, incomplete Freund’s adjuvant was used. Five days after the last immunization, 100 mL of peripheral blood was taken from each animal and immediately supplemented with 5000 IU/mL of heparin to prevent clotting.

### 2.2. Preparation of Phagemid Libraries

Two independent phagemid libraries were constructed for the selection of ALFA-specific binders that can be peptide-eluted at 4 °C. The first library was prepared from total ribonucleic acid (RNA) isolated from peripheral blood mononuclear cells (PBMCs) obtained from fresh alpaca blood. The PBMCs were isolated using Ficoll-Paque PLUS (GE Healthcare, Uppsala, Sweden). Subsequently, total RNA was isolated using a NucleoSpin RNA plus kit (Macherey Nagel, Düren, Germany). The obtained RNA was used for a reverse transcription reaction using Superscript IV Reverse Transcriptase (Thermo Fisher Scientific, Waltham, MA, USA). SdAb-encoding sequences were amplified by a two-step nested polymerase chain reaction (PCR) using the primers CaLl 01 and CaLl 02 [26] and primers F1 and R1 (Appendix A) in a second PCR reaction. The final PCR product was cloned into a pHen2-derived phagemid vector and transformed into TG1 cells, yielding an sdAb library with a complexity of ~2 × 10^8^ individual clones.

For the preparation of the second sdAb library, a structure-guided approach was chosen. Based on the previously published structure of NbALFA bound to an ALFA peptide ([15], PDB: 6I2G), various mutations were introduced to lower the binding affinity of NbALFA and NbALFA^PE^ towards their substrate while ideally retaining their binding specificity. For that, the respective coding sequences were cloned into a pHen2-derived phagemid vector. Then, in both templates, cysteine residue 24 (numbering according to PDB: 6I2G) was mutated to serine. In all four plasmids, all CDR3 residues forming direct contacts with the ALFA peptide were randomized by saturation mutagenesis using degenerate NNK codons, resulting in a library with a theoretical complexity of 6.4 × 10^5^ individual sdAbs. All mutagenesis steps were carried out using primer-directed PCR mutagenesis. The combined pool of mutant phagemids was transformed into TG1 cells, yielding ~1.2 × 10^9^ individual clones.

### 2.3. Selection of Binders—Biopanning

For each library, three subsequent biopanning steps were performed (Figure 1C–E). In each biopanning step, the experimental conditions were adjusted and the stringency, increased. For the first biopanning step, 100 µL of streptavidin-coated MyOne DynaBeads (Thermo Fisher Scientific, Waltham, MA, USA) were loaded with a total of 200 pmol biotinylated target protein (shGFP2-ALFA). The mixture was incubated for 1 h at room temperature, washed three times with 1 mL of phosphate-buffered saline (PBS) and added to a phage suspension containing 2.0 × 10^12^ phages pre-blocked with 20 µg/mL bovine serum albumin (BSA). The suspension was incubated for 1 h at room temperature, transferred to a 4 °C environment and washed once at 4 °C for 30 min with 10 mL of PBS. The remaining binders were eluted at 4 °C with 0.5 mL of PBS containing a 1000-fold excess of ALFA peptide (200 pmol target protein, 200 nmol ALFA peptide). Eluted phages were used to infect *E. coli* TG1 cells for 1 h at 37 °C. The infected TG1 cells were grown overnight at 37 °C. The next day, the culture was diluted and grown to an optical density (OD) of 0.6 before infection with a MK13KO7 helper phage for 1 h at 37 °C. The phages were selected by adding kanamycin (50 µg/mL) and grown overnight at 30 °C. The next day, the phages were purified from the cleared culture supernatants by repeated precipitation with 5% PEG-8000 and 1.5 M NaCl (final concentrations in suspension: 1% PEG-8000 and 300 mM NaCl) and resuspended in a final volume of 4 mL of resuspension buffer. The second biopanning step was performed in a similar way while reducing the amount of target protein (ALFA-biotin) immobilized on the beads to 20 pmol and the elution time to 15 min at 4 °C. For the last biopanning step, 20 pmol of biotin-ALFA was used as a bait. The elution time was reduced to 5 min at 4 °C.

After three rounds of selection, the obtained sdAb-containing libraries were cloned into a pQE-derived expression vector. A total of 96 single clones per library were expressed. Crude lysates were prepared and tested by enzyme-linked immunosorbent assay (ELISA) for their binding properties and their ability to be peptide-eluted at 4 °C. Positive clones were sequenced, aligned and grouped into families. Representative clones were further analyzed biochemically.

### 2.4. Protein Expression and Purification

Recombinant proteins were expressed in *E. coli* from pQE-derived expression vectors (Appendix A). The recombinant proteins ALFA-shGFP2, shGFP2-ALFA, NbALFA, NbALFA^PE^ and NbALFA^CE^ were expressed as N-terminal His_14_-bdSUMO fusion proteins. In general, *E. coli* transformed with the respective plasmids were cultured in terrific broth (TB) medium until an OD of 4.0 was reached. Protein expression was induced by the addition of 0.3 mM isopropyl thiogalactopyranoside (IPTG) at 23 °C, overnight. Before the cells were harvested by centrifugation, the cultures were supplemented with 5 mM ethyleneimine tetraacetate (EDTA) and 1 mM phenylmethylsulfonyl fluoride (PMSF). Subsequently, the cells were lysed in LS buffer (50 mM Tris/HCl, pH 7.5; 300 mM NaCl; 5 mM EDTA) supplemented with 15 mM imidazole/HCl, pH 7.5, and 10 mM dithiothreitol (DTT). The crude lysates were cleared of remaining cellular debris and bound to Ni^2+^-chelate beads for 1 h at 4 °C. The protein-loaded beads were washed extensively, and the proteins were cleaved on column with 100 nM bdSENP1 [27] for 1 h at 4 °C. All proteins were subsequently subjected to size exclusion chromatography. Coupling to an SH-reactive, agarose-based resin was performed using an ectopic cysteine fused to the C-terminus of the respective NbALFA variants via a hydrophilic linker. Product numbers of the resulting Selector resins are summarized in Appendix A.

### 2.5. Off-Rate Assays

For an estimation of the off-rates for ALFA Selector^PE^ and analogous resins coupled to NbALFA^CE^ candidates, 20 µL of substrate-saturated resin was washed four times with PBS and resuspended in 200 µL of PBS containing 200 µM ALFA peptide. At given time points, the progression of elution was quantified by measuring the fluorescence of the GFP-tagged protein released into the supernatant (QBit 3.0; Thermo-Fischer Scientific, Waltham, MA, USA). After the kinetic measurements, all reactions were adjusted to 200 µM peptide concentrations and incubated for 30 min at 30 °C. The fluorescence values obtained after such post-elution were set to 100%. For the experiments shown in Figure 2E,F, each data point represents the average of four independent experiments performed in parallel. The statistical analyses and curve fittings were performed using GraphPad Prism 5.0.

### 2.6. Titration of ALFA Peptide

For each experiment, 250 µL of ALFA Selector^CE^ was saturated with shGFP2-ALFA, extensively washed with PBS containing 0.02% Triton X-100 (TX-100) and packed in a 1 mL syringe equipped with a porous bottom filter frit. For stopped-flow elution at 22 or 4 °C, 70 µL aliquots of PBS + 0.02% TX-100 containing various concentrations of ALFA peptide were added either every 1.5 min (effective flow rate, 0.18 column volumes (CV)/min) or every 5 min (effective flow rate, 0.056 CV/min) and allowed to enter the column by gravity flow while collecting the eluate. In each fraction, the released protein was quantified fluorometrically (QBit 3.0; Thermo-Fischer Scientific, Waltham, MA, USA).

### 2.7. Affinity Purification from E. coli and HeLa Lysates

Affinity-purification experiments with *E. coli* or HeLa lysates were performed with a defined amount of ALFA-tagged target protein. Therefore, cleared *E. coli* lysate or HeLa S100 lysate was blended with 3 µM purified ALFA-tagged target protein (ALFA-shGFP2, shGFP2-ALFA). For each experiment, 25 µL of the indicated ALFA Selector^ST/PE/CE^ (NanoTag Biotechnologies, Göttingen, Germany, Cat No. N1511, N1510, N1512) was incubated with 1 mL of lysate containing the indicated ALFA-tagged target protein for 1 h at room temperature or at 4 °C. After binding, the resins were washed three times in batch with 1 mL of PBS, transferred to a MiniSpin column and washed twice with 0.6 mL of PBS. The resin was then resuspended in 50 µL of PBS containing 200 µM ALFA peptide and incubated for 15 min at room temperature or at 4 °C. All Selector resins were additionally incubated in SDS sample buffer to remove the remaining proteins, heating the samples to 95 °C for 5 min. As specificity controls, the indicated ALFA Selectors were incubated with *E. coli* or HeLa lysate lacking any ALFA-tagged target protein. The samples were resolved by SDS-PAGE and analyzed by Coomassie staining.

### 2.8. Affinity Purification of Low-Abundant Proteins from HeLa Lysates

A volume of 50 mL of HeLa S100 extract was blended with 100 nM shGFP2-ALFA and applied to 1 mL of ALFA Selector^CE^ resin (NanoTag Biotechnologies, Göttingen, Germany, Cat No. N1512) at a flow rate of 0.8–1.0 mL/min. The resin was washed with 10 CV of PBS and eluted by a stepwise addition of PBS containing 1 mM ALFA peptide (250 µL every 3 min) at room temperature. Eluate fractions containing the target protein were pooled. All samples were resolved by SDS-PAGE and further analyzed by Western Blotting. The ALFA-tagged protein was detected using an ALFA-specific HRP-coupled sdAb (NanoTag Biotechnologies, Göttingen, Germany, Cat No. N1501-HRP; Appendix A). Western blotting was essentially performed as described before by Götzke et al. in 2019 [15].

### 2.9. Resistance to Stringent Washing and pH

A volume of 20 µL of ALFA Selector^PE^ or ALFA Selector^CE^ was saturated with either ALFA-shGFP2 or shGFP2-ALFA. The beads were washed four times with Tris-buffered saline (TBS: 25 mM Tris/HCl, pH 7.5; 150 mM NaCl; 2 mM EDTA) and subsequently incubated in 200 µL of the indicated solutions or buffers for 2 h at room temperature with shaking. For post-elution, 250 µM ALFA peptide was added, and the reactions were further incubated for 30 min at 25 °C. The eluates were quantified using a fluorometer (QBit 3.0; Thermo-Fischer Scientific, Waltham, MA, USA) before and after post-elution with ALFA peptide. Pictures were taken upon UV illumination using a Nikon D700 camera equipped with a 105 mm macro lens (Nikon, Minato, Japan).

### 2.10. Regeneration of ALFA Selector^CE^

Regeneration experiments were performed using an Äkta FPLC (Amersham Biosciences, Little Chalfont, UK). A 0.5 mL volume of ALFA Selector^CE^ was packed in a glass column and equilibrated with PBS before saturating it with ALFA-shGFP2. The loaded columns were washed with 10 CV of PBS, and the proteins were eluted in flow (0.1 mL/min) with PBS containing 1 mM ALFA peptide (elution buffer). Eluate fractions containing the target protein were pooled. The protein content was quantified by measuring the absorbance at 280 nm and additionally analyzed by SDS-PAGE. The column was regenerated with 10 CV of glycine pH2.2 (100 mM glycine, pH 2.2; 150 mM NaCl) and re-equilibrated with 10 CV of PBS. After the first regeneration, binding of ALFA-shGFP2 and elution were repeated as described above. The resin was subjected to nine additional regeneration cycles with glycine, before the final target protein loading and elution was performed. All samples were resolved by SDS-PAGE and visualized by Coomassie staining. The experiment was repeated with fresh resin using 100 mM NaOH instead of glycine for regeneration. The elution profiles were plotted using GraphPad Prism 5.0.

## 3. Results

### 3.1. Selection of Cold-elutable ALFA Binders 

The primary aim of this study was to develop an sdAb that would be able to specifically and efficiently capture ALFA-tagged target proteins while being suitable for an efficient competitive elution at 4 °C. We started by creating two independent sdAb libraries (Figure 1). The first sdAb library was obtained from alpacas immunized with ALFA peptides (Figure 1A). The second, semisynthetic sdAb library was constructed using a structure-guided approach (Figure 1B). Here, based on the known structure of substrate-bound NbALFA (PDB: 6I2G), multiple positions within the CDR and scaffold regions of NbALFA were either randomized or mutated (see Materials and Methods). The resulting sdAb libraries were screened by a customized, off-rate-driven phage display using cycles of binding, stringent washing and competitive peptide elution at 4 °C (Figure 1C–E). The selection stringency was successively increased by reducing the time allowed for competitive phage elution from an initial 30 min to only 5 min in cycle 3. Using this approach, we ensured an enrichment of sdAbs that stably associated with ALFA-tagged proteins while still allowing for an efficient release at 4 °C. After three panning cycles, the NbALFA^CE^ candidates were cloned into a prokaryotic expression vector, expressed in *E. coli* (Figure 1F) and analyzed by ELISA for the desired properties (Figure 1G,H). From the 192 clones analyzed, nine nonredundant NbALFA^CE^ candidates representing the most dominant sdAb families were chosen for further characterization.

### 3.2. Novel ALFA Binders Facilitate Competitive Elution of ALFA-Tagged Target Proteins at 4 °C

All selected NbALFA^CE^ candidates were recombinantly expressed in *E. coli*, purified and immobilized on an agarose-based resin. For an initial characterization, each resin was loaded with a green fluorescent protein variant (shGFP2) [28] fused to a single ALFA-tag (shGFP2-ALFA) and used as a visible model substrate to follow elution on ice upon addition of an excess of ALFA peptide. The binding and the elution kinetics were semiquantitatively assessed by visual inspection (Table 1).

While all the candidates were able to capture shGFP2-ALFA, the observed dissociation kinetics differed considerably. When compared to NbALFA^PE^, which was used as a control and is known to only slowly release target proteins at low temperature, the elution from all NbALFA^CE^ candidate clones was much more efficient. The NbALFA^CE^ candidates C1, E10, A12 and F10 showed efficient elution of shGFP2-ALFA within an optimal timeframe of 5–10 min and were selected for a more detailed analysis. Clones with slower or faster elution kinetics were excluded from all further analyses.

For a more thorough characterization, agarose resins coupled to the NbALFA^CE^ candidates C1, E10, A12 and F10 were loaded with shGFP2 substrates harboring an N- or C-terminal ALFA-tag, respectively. After washing, ALFA peptide was added in excess, and the elution kinetics at 4 °C were followed by measuring the fluorescent material released into the supernatant (Figure 2). The amount of ALFA peptide used in this “batch elution mode” (200 µM peptide and 10 resin bed volumes) corresponded to an ~15-fold excess over the available binding sites, which is sufficient to trigger an efficient competitive elution. For all the resins addressed, the elution kinetics of shGFP2-ALFA were consistent with the rough evaluations conducted before (Table 1). However, striking differences were observed when comparing the elution of shGFP2 harboring N- or C-terminal ALFA-tags: both resins coupled to the NbALFA^CE^ candidates C1 and E10 originating from the semisynthetic library showed very similar and consistent elution kinetics for substrates tagged at either terminus (Figure 2A,B). By contrast, the elution kinetics of the two substrates differed strikingly when analyzing the resins coupled to the NbALFA^CE^ candidates originating from the immune library (Figure 2C,D). As an NbALFA^CE^ candidate should ideally recognize ALFA-tagged proteins irrespective of the localization of the ALFA-tag within the target protein, we excluded both NbALFA^CE^ candidates originating from the immune library. Due to its slightly stronger binding to both ALFA-tagged substrates, we finally selected the candidate C1 as our final NbALFA^CE^. This clone combines a robust binding of ALFA-tagged substrates with a sufficiently fast and consistent elution kinetics at 4 °C irrespective of the localization of the ALFA-tag within the target protein. The agarose-based resin featuring immobilized NbALFA^CE^ was called ALFA Selector^CE^.

When directly comparing the elution kinetics of ALFA Selector^PE^ and ALFA Selector^CE^ at 4 and 25 °C, major differences could be observed (Figure 2E,F). While competitive elution was possible within 15–20 min at 4 °C or 3 min at 25 °C when using ALFA Selector^CE^, efficient elution from ALFA Selector^PE^ could only be achieved at 25 °C. For both resins, any spontaneous leakage of bound target protein in the absence of peptide was negligible.

### 3.3. Impact of ALFA Peptide Concentration on Elution

Having established that ALFA-tagged proteins can be eluted efficiently from ALFA Selector^CE^ at 4 °C, the impact of the ALFA peptide concentration on the elution of an ALFA-tagged target protein was tested. In a first set of experiments, ALFA Selector^CE^ saturated with shGFP2-ALFA was subjected to competitive elution with varying ALFA peptide concentrations at 4 °C and room temperature (~22 °C) in a “stopped-flow” setup (Appendix A). In this elution mode, the elution buffer is added intermittently in small portions, resulting in a low net flow rate. In a third experiment, competitive elution was performed at 22 °C in “gravity-flow” mode, resulting in a higher net flow rate. Under all the conditions tested, the peak protein concentration and the sharpness of the elution peak increased upon raising the peptide concentration. When using 1 mM ALFA peptide during elution, >90–95% of the target protein could be recovered within 3–4 fractions (0.84–1.12 column volumes), indicating that efficient elution is possible at 4 °C in “stopped-flow” mode (Appendix A) and at 22 °C even in flow mode (Appendix A). Even higher peptide concentrations led to only marginally better results.

### 3.4. Affinity Purification Using Different ALFA Selector Variants

We next wanted to compare the features of the well-characterized high-affinity ALFA Selector^ST^ and intermediate-affinity ALFA Selector^PE^ [15] with the novel ALFA Selector^CE^ in single-step affinity purification experiments using shGFP2-ALFA at 22 and 4 °C (Figure 3A,B). As expected, all the resins specifically bound the ALFA-tagged target protein at both temperatures with comparable capacity. Consistent with our previous observations [15], the competitive peptide elution from ALFA Selector^ST^ was generally inefficient at all the tested temperatures. Thus, the bound proteins had to be post-eluted with SDS sample buffer (Figure 3A,B, left panel). By contrast, the bound target protein could be efficiently released by peptide elution under native conditions at 22 °C from both ALFA Selector^PE^ and ALFA Selector^CE^ (Figure 3A,B, middle and right panels). When comparing the efficiency of peptide elution at 4 °C, differences became apparent between ALFA Selector^PE^ and ALFA Selector^CE^: while the peptide elution from ALFA Selector^PE^ was inefficient, the target protein was readily released from ALFA Selector^CE^ under identical conditions (Figure 3B, middle and right panels).

### 3.5. Affinity Purification of Low-Abundant Proteins

The expression levels of the target proteins in various experimental systems can vary significantly. In order to analyze if ALFA Selector^CE^ can also capture low-abundant ALFA-tagged proteins, we applied 50 mL of a HeLa lysate containing only 100 nM ALFA-shGFP2 to 1 mL of ALFA Selector^CE^ resin using gravity flow (flow rate, ~1 mL/min). After washing, the column was eluted “in flow” with 1 mM ALFA peptide in PBS. Under these experimental conditions, a nearly complete binding of the ALFA-tagged substrate to the resin was achieved (Figure 3C; Appendix A). Only after prolonged exposure, trace amounts of the target protein could be detected by Western blotting in the non-bound fraction. The affinity of ALFA Selector^CE^ is therefore sufficient for an efficient capture of ALFA-tagged proteins, even from dilute samples.

### 3.6. Buffer Compatibility

To compare the behavior of ALFA Selector^PE^ and ALFA Selector^CE^ in different buffers, both resins were first loaded with either shGFP2-ALFA or ALFA-shGFP2. After thorough washing with PBS, the resins were transferred into various buffers and analyzed after shaking for 2 h at room temperature (Figure 4). In general, both N- and C-terminally ALFA-tagged shGFP2 showed a remarkably stable association with both ALFA Selector resins, even in the presence of high salt concentrations (up to >3 M NaCl), nondenaturing detergents, and small-to-moderate amounts of urea. By contrast, denaturing detergents such as SDS and urea concentrations >3 M led to a substantial release of the target protein into the supernatant. For both resins, the substrate leakage was slightly more pronounced when using ALFA-shGFP2 as a substrate instead of shGFP2-ALFA (Figure 4, lower and upper panels). In line with the weaker interaction expected for the novel cold-elutable ALFA Selector^CE^, this resin exhibited a higher tendency to release target proteins into the supernatant than ALFA Selector^PE^ (compare Figure 4A,B).

### 3.7. Regeneration of ALFA Selector^CE^

Finally, we analyzed if it was possible to regenerate ALFA Selector^CE^ after use. Therefore, the resin was loaded with saturating amounts of ALFA-shGFP2, washed and eluted in flow mode with PBS containing 1 mM ALFA peptide. After 1 and 10 cycles of regeneration under acidic or basic conditions (100 mM glycine, pH 2.2, and 150 mM NaCl, or 100 mM NaOH, respectively), the same sequence of loading, washing and peptide elution was performed again. The absorption profiles (measured at 280 nm) for the complete regeneration experiments performed under acidic (Figure 5A) and basic conditions (Appendix A) showed only minor variations during the loading and elution of the ALFA-tagged target protein before regeneration and after 1 or 10 regeneration cycles. Similar amounts and quality of the eluted target protein were also observed when analyzing the respective eluates by SDS-PAGE and Coomassie staining (Figure 5B and Appendix A), indicating that there is no obvious loss in binding capacity even after 10 regeneration cycles. In line with this observation, affinity purification experiments using the regenerated resins showed that the overall capacity and the low nonspecific background binding to ALFA Selector^CE^ remained largely unchanged (Figure 5C and Appendix A). ALFA Selector^CE^ can therefore be used multiple times without a substantial loss of performance.

## 4. Discussion

We recently described the ALFA system comprising the ALFA-tag, a novel epitope tag and a set of ALFA-specific sdAbs, NbALFA and NbALFA^PE^ [15]. NbALFA shows an extremely stable and robust association with ALFA-tagged proteins and is ideally suited for in vivo protein manipulations, highly sensitive affinity capturing and advanced detection and imaging techniques. By contrast, NbALFA^PE^ was optimized for competitive elution at ambient temperature, and can be used for the purification of proteins or demanding living cell isolation under physiological conditions [15]. To broaden the versatility of the ALFA system even further, we, here, aimed at developing NbALFA^CE^, a new sdAb combining high selectivity and stable binding of ALFA-tagged proteins with an efficient competitive elution at 4 °C under physiological buffer conditions.

To find sdAbs fulfilling these virtually diverging goals, we started two independent phage-display screens based on different libraries. The first library was created by conventional immunization of alpacas, while the second library was designed by introducing mutations into NbALFA and NbALFA^PE^ by a rational, structure-guided approach. After applying stringent selection criteria, both libraries yielded NbALFA^CE^ candidates that allowed stable binding of ALFA-tagged proteins while permitting an efficient competitive elution at 4 °C. In addition to these requested features we could directly screen for, optimal NbALFA^CE^ candidates should ideally show comparable binding strength and elution kinetics for target proteins harboring N- or C-terminal ALFA-tags.

Surprisingly, however, we noticed striking differences depending on the origin of the clones analyzed. Both NbALFA^CE^ candidates derived from the immunized library showed a strong preference for proteins harboring a C-terminal ALFA-tag. By contrast, clones originating from the semisynthetic library showed remarkably similar elution kinetics irrespective of the localization of the ALFA-tag within the target protein—just as described for NbALFA and NbALFA^PE^, indicating that the NbALFA^CE^ candidates originating from the semisynthetic library preserved this feature.

The consistent recognition of target proteins tagged at either terminus is, therefore, not necessarily self-evident. By generating a semisynthetic library based on sdAbs that already possess this requested feature, we aimed to increase the chances of also finding these properties in the descendant clones. We believe that sdAb clones with similar properties may also be present in the immunized library. However, it is hard to envision a selection procedure that would result in sdAbs featuring these properties in a stringent and reproducible manner. Therefore, once a well-characterized sdAb displaying some of the desired features and biophysical properties is available, using this sdAb as the precursor of a new library seems to be an effective alternative to proceeding with a conventional de novo discovery process. In analogy, this general approach can potentially be applied to finetune certain binding properties (e.g., binding strength) while making sure the sdAb keeps other desired properties, such as its biochemical properties, its charge distribution or its specificity. Such an approach is most promising if a direct screening for a rare feature is not feasible or would require unreasonable effort.

The final NbALFA^CE^ displays four modifications with respect to NbALFA. In comparison to the NbALFA^PE^, featuring intermediate affinity, the off-rate of NbALFA^CE^ is substantially enhanced, which is a prerequisite for achieving an efficient competitive elution at 4 °C (Figure 2E,F). The resin can, nevertheless, efficiently capture ALFA-tagged proteins even from dilute samples using a simple gravity-flow purification system (Figure 3C). As expected, the higher off-rate also influences the biochemical stability of the complexes formed with ALFA-tagged proteins. NbALFA^CE^ is, therefore, slightly more sensitive than NbALFA^PE^ towards denaturing reagents such as urea or guanidinium hydrochloride and certain detergents (Figure 4). However, under physiological buffer conditions or even at drastically elevated salt concentrations, any spontaneous leakage of target protein from an NbALFA^CE^-coupled resin (ALFA Selector^CE^) is negligible (Figure 2E,F and Figure 4).

Although the N- and C-terminal ALFA-tagged proteins behaved in a strikingly similar way in the kinetic off-rate assays performed with ALFA Selector^CE^, we observed subtle differences regarding their biochemical properties and pull-down behavior, especially under nonphysiological conditions (Figure 4).The observed differences in biochemical behavior are marginal for our model substrate shGFP2. However, as for any tag, it may be advisable to test the optimal position of the tag on other proteins or protein complexes, as steric effects or target-specific surface properties might influence its accessibility.

When varying the concentration of the ALFA peptide during elution from ALFA Selector^CE^, we observed the sharpest elution profiles in stopped-flow and flow mode if 1 mM peptide was used. At this concentration, the excess of peptide over resin-bound sdAb is sufficiently high to ensure a nearly off-rate-limited elution. Therefore, higher peptide concentrations will not increase the elution speed further. When using larger elution volumes, as generally performed in “batch” mode, peptide concentrations of only 200 µM are fully sufficient (Figure 2), as a similar excess of peptide over the resin-bound sdAb is achieved. While competitive peptide elution at 4 °C from ALFA Selector^CE^ requires 15–20 min for completion, elution at room temperature can even be performed using gravity flow (Figure 3C, Appendix A and Figure 5).

At all tested temperatures, rapid and complete elution from ALFA Selector^CE^ can also be achieved by applying acidic or basic conditions (Figure 5 and Appendix A), which was exploited to perform cleaning-in-place (CIP) regeneration. After 10 cycles of regeneration, the binding capacity and specificity of ALFA Selector^CE^ remained largely unaffected (Figure 5B,C and Appendix A). ALFA Selector^CE^ can therefore be reused several times without a loss of performance.

In summary, by introducing the novel, cold-elutable NbALFA^CE^, the same ALFA-tagged protein can be bound by three alternative NbALFA variants featuring different affinities, thereby increasing the versatility of the ALFA system even further. ALFA Selector^CE^ combines a highly specific and biochemically stable binding of ALFA-tagged proteins with the ability to readily elute at low temperatures under physiological buffer conditions. Thereby, the newly characterized ALFA Selector^CE^ is an ideal tool for the clean, single-step purification of temperature-labile targets and applications requiring perfect structural and functional conservation, such as for cryo-electron microscopy or the purification of sensitive enzymes, respectively.

## Figures and Tables

**Figure 1 biomolecules-11-00269-f001:**
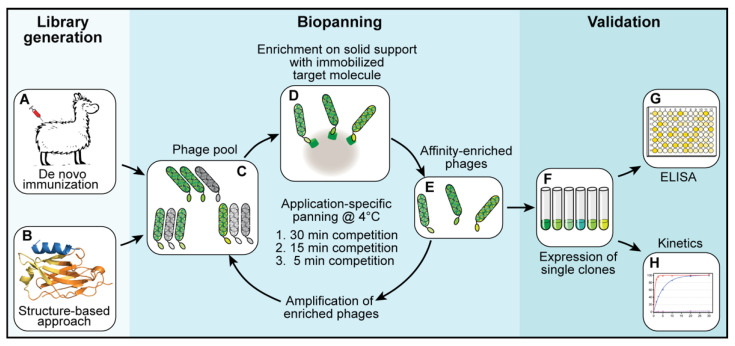
Schematic depiction of the workflow followed to create cold-elutable ALFA binders. Two separate sdAb libraries were generated by either de novo immunization of alpacas (**A**) or a structure-guided approach based on the known structure of NbALFA in complex with the ALFA peptide (**B**). The resulting phage pools were separately used as input material for phage displays (**C**–**E**). In three rounds of biopanning, the selection stringency was increased by reducing both the amount of immobilized target protein and the time allowed for competitive elution at 4 °C. Eluted phages were amplified and used as input for the next round of biopanning. After three rounds of biopanning, enriched NbALFA^CE^ candidates were subcloned and expressed in *E. coli* (**F**). Candidates were further assayed by ELISA for their binding and elution properties (**G**) or subjected to a kinetic characterization (**H**).

**Figure 2 biomolecules-11-00269-f002:**
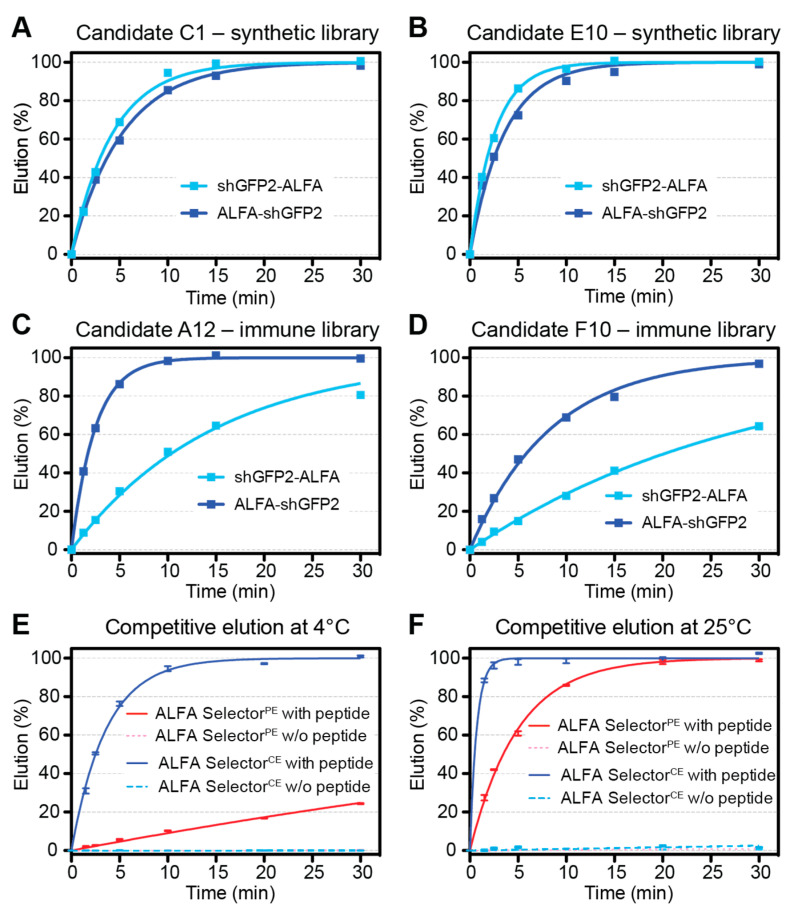
Kinetic characterization of NbALFA^CE^ candidate clones. (**A**–**D**): Selected NbALFA^CE^ candidates were immobilized on an agarose-based resin. All resins were charged with either shGFP2-ALFA (light blue curves) or ALFA-shGFP2 (dark blue curves). Upon competitive elution in batch mode using 200 µM ALFA peptide at 4 °C, target protein elution was quantified by measuring the fluorescent material released into the supernatant. (**E**,**F**): ALFA Selector^PE^ (PE, red curves) and ALFA Selector^CE^ based on candidate clone C1 (CE, blue curves) were charged with shGFP2-ALFA and subjected to competitive peptide elution at 4 °C (**E**) or 25 °C (**F**), respectively. Parallel reactions without addition of peptide served as controls (dotted red and dotted blue curves). Elution was quantified as described for panels (**A**–**D**). The data points show mean values from four independent experiments including standard deviations. Curves represent fits to first-order association kinetics.

**Figure 3 biomolecules-11-00269-f003:**
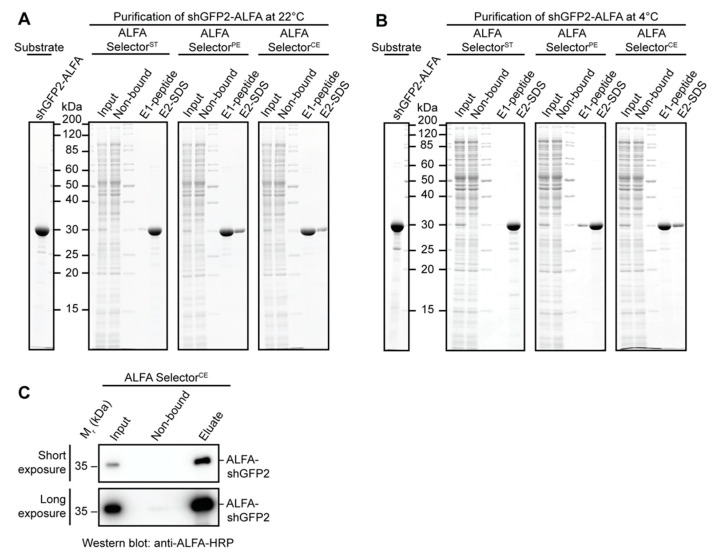
Affinity purification of ALFA-tagged proteins from complex lysates. (**A**,**B**): An *E. coli* lysate was blended with 3 µM recombinant purified shGFP2-ALFA (**A**,**B**) and incubated with ALFA Selector^ST^, ALFA Selector^PE^ or ALFA Selector^CE^. After washing, charged resins were eluted with 200 µM ALFA peptide at 22 °C (**A**) or 4 °C (**B**) (E1-peptide). For all Selectors, remaining proteins were subsequently eluted with hot SDS sample buffer (E2-SDS). Amounts loaded in the SDS-PAGE for all eluate fractions correspond to the material eluted from 1 µL of Selector resin. Images show gels after staining with Coomassie Brilliant Blue. (**C**): One-step affinity purification of low-abundant proteins using ALFA Selector^CE^: 50 mL of HeLa lysate containing 100 nM ALFA-shGFP2 was passed over 1 mL of ALFA Selector^CE^ at room temperature using gravity flow. After washing, the column was eluted with PBS containing 1 mM ALFA peptide. Fractions were pooled and analyzed by SDS-PAGE and Western blotting. Amounts loaded correspond to 1/20,000 of the input and flow-through material and 1/2000 of the eluate. Shown is a representative blot after short (upper panel) and long exposure (lower panel). Full blots are shown in Appendix A.

**Figure 4 biomolecules-11-00269-f004:**
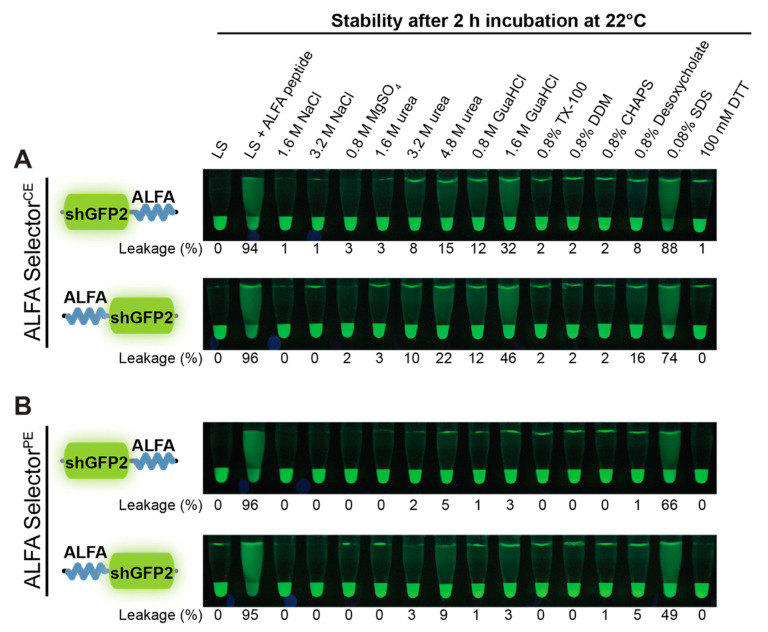
Buffer compatibility. ALFA Selector^CE^ (**A**) and ALFA Selector^PE^ (**B**) were saturated with shGFP2-ALFA (upper panels) or ALFA-shGFP2 (lower panels), washed extensively with PBS and incubated in a 10-fold volume of the indicated substances for 2 h at 22 °C. The leakage of target protein from the Selector resin was analyzed by quantifying the fluorescent material released into the supernatant before (shown here) and after post-elution with ALFA peptide (control shown in Appendix A). The degree of leakage/dissociation of target protein from the beads is given as % of total.

**Figure 5 biomolecules-11-00269-f005:**
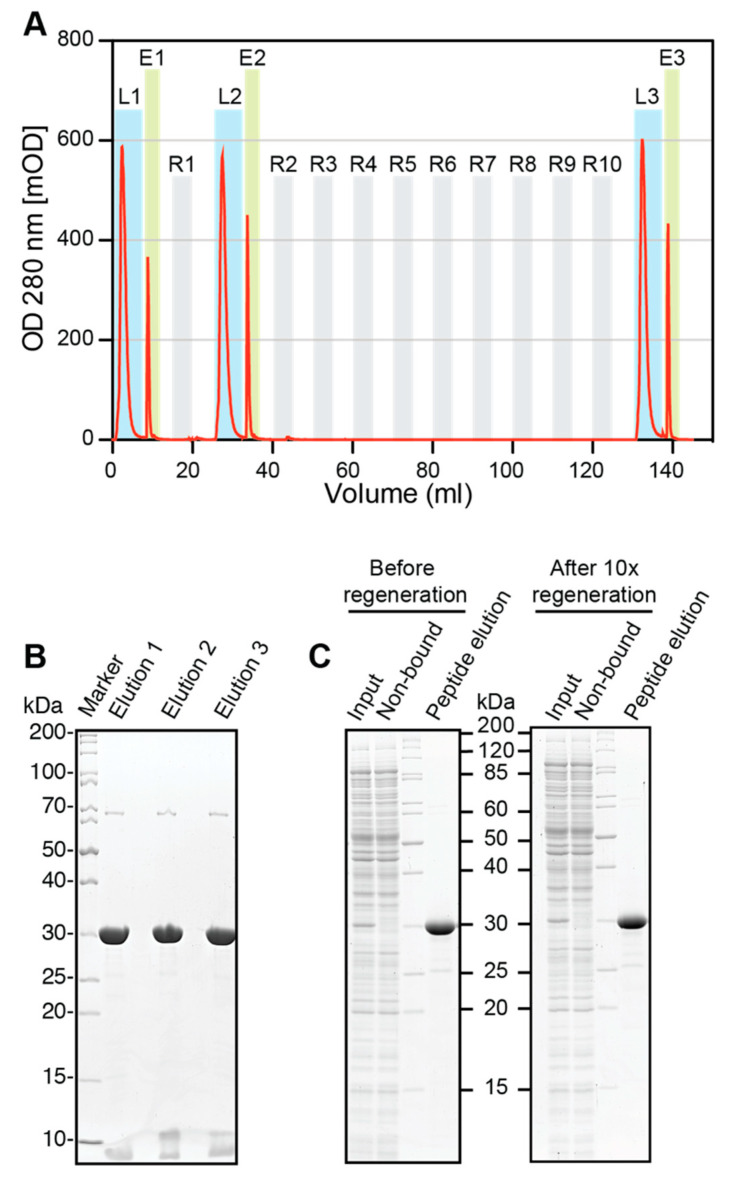
Regeneration of ALFA Selector^CE^ under acidic conditions. (**A**): 0.5 mL of ALFA Selector^CE^ was subjected to a first cycle of loading with ALFA-shGFP2 (L1) and competitive peptide elution (E1). The column was regenerated under acidic conditions using 100 mM glycine and 150 mM NaCl at pH 2.2 (R1) and re-equilibrated with PBS before starting a second cycle of loading and elution (L2 and E2). After 9 additional repeated regeneration/re-equilibration steps (R2–R10), the column was loaded and eluted a third time (L3, E3). The whole procedure was followed by recording the optical density at 280 nm (OD280; red curves). (**B**): Fractions for each elution step were collected, quantified and analyzed by SDS-PAGE and Coomassie staining. (**C**): Effect of regeneration on nonspecific background binding. Single-step affinity purification from *E. coli* lysate blended with shGFP2-ALFA was performed using ALFA Selector^CE^ either before regeneration or after 10 cycles of regeneration with glycine, pH 2.2 (left and right panels, respectively). The experiment was essentially performed as described for Figure 3A,B.

**Table 1 biomolecules-11-00269-t001:** Binding properties of selected ALFA binders obtained from an immune library and a semisynthetic library.

NbALFA^CE^ Candidate on Resin		Remaining Signal after Competitive Elution on Ice				
Clone	Origin of Library	0 min	2 min	5 min	10 min	30 min
C1	Semisynthetic	+++	+	(+)	-	-
D3	Semisynthetic	+++	(+)	-	-	-
E5	Semisynthetic	+++	(+)	-	-	-
E10	Semisynthetic	+++	++	+	(+)	-
H11	Semisynthetic	+++	(+)	-	-	-
A12	Immune library	+++	+	(+)	-	-
C6	Immune library	+++	+	-	-	-
F9	Immune library	+++	++	+	(+)	(+)
F10	Immune library	+++	++	+	(+)	-
NbALFA^PE^ control		+++	+++	+++	++	+

Legend: +++: target protein completely bound to Selector resin; ++: moderate decrease in signal after competitive elution; +: strong decrease in signal after competitive elution; (+): barely measurable signal after competitive elution; -: no signal remaining on resin.

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
