# Peer review of "Discovery and Characterization of an ALFA-Tag-Specific Affinity Resin Optimized for Protein Purification at Low Temperatures in Physiological Buffer"

_biomolecules, 2021, doi:10.3390/biom11020269_

Round 1

Reviewer 1 Report

The manuscript describes a panning methodology for the identification of a nanobody with specific binding affinity towards its antigen. More specifically, the authors identified a mutant with kinetic characteristics suitable for reversible elution at 4°C of proteins fused to ALFA-tag. The adopted panning strategy is interesting for itself and the increasing use of ALFA-tag and its applications in the research community justifies the efforts for optimizing the reagents related to this technology. The manuscript is well conceived and the experiments reported in the Results are rationally organized and performed correctly. In few occasions some procedures are not particularly clear and it would be convenient to anticipate pieces of information that are indeed available in the Discussion. The major limitation of the contribution is the absence of information relatively to the rational mutation strategy that led to the semi-synthetic library design. The reviewer is aware of the probably existing necessity to protect commercial interests, therefore let to the Editor the decision if the lack of data availability is compatible with the standards of scientific publications.

Minor points:

  • L157: What was the chemistry used to bind the Nbs to the beads?
  • L301: it is not clear why C1 was preferred to E10: did the authors considered the KD?
  • L302: C1 was selected following the elution kinetics of the alternative constructs of a single protein? I’d find it extremely risky, given the fact that any passenger protein could possess its specific folding and different availability of the N- and C-terms. The authors discuss the point (L491-494) but could maybe better add a Suppl. Fig. with another example, if available
  • 2E&F. The figures are not self-evident. Maybe the indications “SelectorPE” and CE should appear instead of “4°C PE”. In the legend maybe useful to repeat that SelectorCE “corresponds” to C1.
  • L359: why 200 uM ALFA peptide was used when just above 1 mM was indicated as optimal concentration? This point is discussed at L500, but it would be useful to better clarify in the Results the peculiarities of the different elution strategies
  • Fig 3C: maybe “SelectorCE” should be indicated in the figure, given the fact that there is no symmetry with the experiments reported in A and B (in which the three resins were compared)
  • L375: the elution volume and the concentration should be indicated for comparison with the input
  • L418-419: “significant…significantly”. Avoid repeats and even consider to change both terms to avoid the criticism that “significant” is used, in scientific language, to statistical evaluations that are missing in the present case
  • Figure 5C. The quality of the figure should be substantially improved and the corresponding experiment better described in both the text and the legend.
  • L476: the mutations are not reported (see the section with general comments)

Reviewer 2 Report

The ALFA tag is novel affinity tag with very high affinity for a specific sdAb that the authors had previously described. The original sdAb bound so strongly to the ALFA tag that it was difficult to compete with ALFA peptide, limiting the use of the ALFA-tag system for recombinant protein purification akin to the widely used systems, such as Strep- or HisTag. Using a clever panning strategy, the authors now describe ALFA-specific sdAbs that have reduced affinity that allows for competition at 4 degC and complete elution of an ALFA-tagged GFP substrate from sdAb-coupled agarose. The authors characterize this new resin, called ALFA-Selector CE, thoroughly and compare and contrast it with the other two resins their company sells. The experiments are extremely well carried out, extremely well presented and written, and highly convincing.

This is a very important study that describes a new tool for affinity purification. Because the authors show that the new resin can easily be regenerated, it may well substitute Ni-NTA as the default affinity chromatography step. 

We have no significant criticism. This paper should be published immediately.

minor comment:

1.)

Line 154 – 100mM DTT – is that a typo?

2.)

In multiple places NbALFACE is unsightly hyphenated to NbAL-FACE. Please avoid for readability.
